# The Differences in the Prevalence of Cardiovascular Disease, Its Risk Factors, and Achievement of Therapeutic Goals among Urban and Rural Primary Care Patients in Poland: Results from the LIPIDOGRAM 2015 Study

**DOI:** 10.3390/jcm10235656

**Published:** 2021-11-30

**Authors:** Krzysztof Studziński, Tomasz Tomasik, Adam Windak, Maciej Banach, Ewa Wójtowicz, Mirosław Mastej, Maciej Tomaszewski, Dimitri P. Mikhailidis, Peter P. Toth, Alberico Catapano, Kausik K. Ray, George Howard, Gregory Y.H. Lip, Fadi J. Charchar, Naveed Sattar, Bryan Williams, Thomas M. MacDonald, Peter E. Penson, Jacek J. Jóźwiak

**Affiliations:** 1Department of Family Medicine, Jagiellonian University Medical College, Bochenska 4, 31-061 Krakow, Poland; krzysztof.studzinski@uj.edu.pl (K.S.); mmwindak@cyf-kr.edu.pl (A.W.); ewa.woj@poczta.onet.pl (E.W.); 2Polish Mothers Memorial Hospital Research Institute, 93-338 Lodz, Poland; maciejbanach77@gmail.com; 3Department of Hypertension, Chair of Nephrology and Hypertension, Medical University of Lodz, 90-136 Lodz, Poland; 4Cardiovascular Research Centre, University of Zielona Gora, 65-417 Zielona Gora, Poland; 5Mastej Medical Center, 38-200 Jasło, Poland; miroslaw@mastej.pl; 6Division of Cardiovascular Sciences, Faculty of Biology, Medicine and Health, University of Manchester, Manchester M13 9PT, UK; maciej.tomaszewski@manchester.ac.uk; 7Department of Clinical Biochemistry, Royal Free Hospital, University College London, London NW3 2QG, UK; mikhailidis@aol.com; 8Cicarrone Center for the Prevention of Cardiovascular Disease, Johns Hopkins University School of Medicine, Baltimore, MD 21287, USA; peter.toth@cghmc.com; 9CGH Medical Center, Sterling, IL 61081, USA; 10Department of Pharmacological Sciences, University of Milano and Multimedica IRCCS, 20099 Milano, Italy; alberico.catapano@unimi.it; 11Department of Primary Care and Public Health, Imperial Centre for Cardiovascular Disease Prevention, Imperial College, Kensington, London W6 8RP, UK; k.ray@imperial.ac.uk; 12Department of Biostatistics, School of Public Health of Alabama at Birmingham, Birmingham B15 2TT, UK; ghoward@uab.edu; 13Liverpool Centre for Cardiovascular Science, University of Liverpool and Liverpool Heart & Chest Hospital, Liverpool L14 3PE, UK; gregory.lip@liverpool.ac.uk; 14Aalborg Thrombosis Research Unit, Department of Clinical Medicine, Aalborg University, 9000 Aalborg, Denmark; 15School of Health and Life Sciences, Federation University Australia, Ballarat, VIC 3350, Australia; f.charchar@federation.edu.au; 16Institute of Cardiovascular and Medical Science, University of Glasgow, Glasgow G12 8TA, UK; naveed.sattar@glasgow.ac.uk; 17NIHR University College London Biomedical Research Centre, University College London and University, London NW1 2BU, UK; bryan.williams@ucl.ac.uk; 18College London Hospitals NHS Foundation Trust, London NW1 2BU, UK; 19MEMO Research, Ninewells Hospital and Medical School, University of Dundee, Dundee DD1 9SY, UK; t.m.macdonald@dundee.ac.uk; 20School of Pharmacy and Biomolecular Sciences, Liverpool John Moores University, Liverpool L2 2QP, UK; p.penson@lmju.ac.uk; 21Liverpool Centre for Cardiovascular Science, Liverpool L69 7TX, UK; 22Department of Family Medicine and Public Health, Institute of Medical Sciences, University of Opole, Oleska 48, 45-052 Opole, Poland; jacek.jozwiak@uni.opole.pl

**Keywords:** primary health care, cardiovascular diseases, risk factors, urban, rural, Poland

## Abstract

A nationwide cross-sectional study, LIPIDOGRAM2015, was carried out in Poland in the years 2015 and 2016. A total of 438 primary care physicians enrolled 13,724 adult patients that sought medical care in primary health care practices. The prevalence of hypertension, diabetes mellitus, dyslipidaemia, and CVD were similar in urban and rural areas (49.5 vs. 49.4%; 13.7 vs. 13.1%; 84.2 vs. 85.2%; 14.4 vs. 14.2%, respectively). The prevalence of obesity (32.3 vs. 37.5%, *p* < 0.01) and excessive waist circumference (77.5 vs. 80.7%, *p* < 0.01), as well as abdominal obesity (43.2 vs. 46.4%, *p* < 0.01), were higher in rural areas in both genders. Mean levels of LDL-C (128 vs. 130 mg/dL, *p* = 0.04) and non-HDL-C (147 vs. 148 mg/dL, *p* = 0.03) were slightly higher in rural populations. Altogether, 14.3% of patients with CVD from urban areas and 11.3% from rural areas reached LDL <70 mg/dL (*p* = 0.04). There were no important differences in the prevalence of hypertension, diabetes, dyslipidaemia, and CVD, or in mean levels of blood pressure, cholesterol fractions, glucose, and HbA1c between Polish urban and rural primary care patient populations. A high proportion of patients in cities and an even-higher proportion in rural areas did not reach the recommended targets for blood pressure, LDL-C, and HbA1c, indicating the need for novel CVD-prevention programs.

## 1. Introduction

Differences in healthcare access, utilisation, and health status between rural and urban populations are described in the medical literature [1,2]. Cardiovascular diseases (CVD) are the leading causes of death and disability globally [3]. Well-known cardiovascular (CV) risk factors include elevated blood pressure (BP), dyslipidaemia, hyperglycaemia, obesity, and smoking. The prevalence of CVD and CV risk factors is high in Central and Eastern European (CEE) countries [4]. Currently, in Poland, over 15% of the patients attending primary health practices are diagnosed with CVD, over 60% have hypertension (HTN), over 80% are dyslipidemic, and approximately 15% of patients are diagnosed with diabetes mellitus (DM) [5].

Differences in CVD prevalence and levels of CV risk factors between urban and rural regions have been confirmed all across Europe [6,7,8,9]. Additionally, the Prospective Urban Rural Epidemiology (PURE) Study (which enrolled 156,424 persons from 3 high-income, 10 middle-income, and 4 low-income countries) reported higher rates of CV events in rural than urban areas in middle-income and low-income countries. By comparison, there were no such differences in high-income countries. However, at the same time, the prevalence of the CV risk factors was higher in urban than in rural communities [10].

Despite the high prevalence of CV risk factors, achieving the recommended levels of cholesterol fractions, blood pressure, and glucose remains challenging. The authors of the DA VINCI study revealed that 54% of patients achieved their risk-based LDL-C goals [11]. Meanwhile, data from the TERCET registry indicate that after ST-elevated myocardial infarction (STEMI), 32.4% of patients achieved LDL-C < 70 mg/dL, compared to 29.9% patients after a non-ST elevated myocardial infarction (NSTEMI) [12].

Due to health inequities between urban and rural regions, place of residence remains one of the essential determinants of health [9]. Nevertheless, it is not currently known whether urban–rural differences exist in the prevalence of CV risk factors and the achievement of therapeutic goals in Poland and some other CEE countries.

The aim of this study was to analyse the differences between the patients attending primary health care (PHC) practices located in urban and rural settings in Poland in reference to (1) the prevalence of CV risk factors and CVD and (2) the proportion of individuals achieving therapeutic goals in the treatment of HTN, dyslipidaemia, and DM.

## 2. Materials and Methods

We conducted a secondary analysis of the data from the nationwide cross-sectional study LIPIDOGRAM2015, which was carried out in Poland in the fourth quarter of 2015 and first and second quarters of 2016 [13]. A group of 438 physician-investigators in 398 practices were selected in a manner proportional to the number of inhabitants in 16 Polish major administrative regions. The study actively enrolled 13,724 adult patients seeking medical care for any medical reason in PHC practices. Details of sampling, selection, and measurement methods have been described previously [5,13]. Briefly, for each participant in the study, the physician-investigator completed a 28-item questionnaire, which gathered basic demographic (age, gender, level of education, place of residence) and clinical (chronic diseases, lifestyle habits including diet, physical activity, smoking status, family history of CVD, and current pharmacotherapy) data. Anthropometric measurements were made (height, body weight, waist circumference, and hip circumference) and the body mass index (BMI) was calculated. On the same day, measurements of BP, heart rate, and fasting glucose were taken. Blood samples were collected for measurement of the lipid profile. Glycated haemoglobin was assessed in a subgroup of patients (random sample of up to four patients in each PHC practice).

### 2.1. Definitions and Classification

HTN, DM, and dyslipidaemia were defined according to the relevant guidelines that were in force in 2015 [14,15,16].

Participants were classified as having HTN, DM, and dyslipidaemia when: (1) those diseases were previously diagnosed; or (2) patient used at least one hypotensive, lipid-lowering or hypoglycaemic medication, respectively. Participants were also defined as having HTN when office SBP was ≥140 mmHg and/or office DBP was ≥90 mmHg (average from two measurements). Participants were also defined as having dyslipidaemia when the measured LDL-C level was ≥115 mg/dL and/or TG level was ≥150 mg/dL.

The presence of coronary heart disease, previous myocardial infarction, atrial fibrillation, ischemic or haemorrhagic stroke, and medications the patients used were reported in the questionnaire by the physician who cared for the patient.

We used the WHO classification of obesity, based on BMI [17] and the norms of the International Diabetes Federation (IDF) for waist circumference (<80 cm in women and <94 cm in men) [18]. Overweight was defined as BMI 25.0–29.9 and obesity ≥30.0, respectively. We defined abdominal obesity as a waist–hip ratio (WHR) ≥ 1 for men and WHR ≥ 0.85 for women [19].

Patients who smoked at least one cigarette per day on the day of inclusion in the study were classed as current smokers. Patients who had smoked at least 100 cigarettes in their lifetime, quit smoking, and did not smoke on the day of inclusion in the study were classed as ex-smokers. Patients who never smoked cigarettes were classed as non-smokers.

Therapeutic goals for HTN (SBP < 140 mmHg and DBP < 90 mmHg), DM (HbA1c < 7%), and dyslipidaemia (patients with CVD: LDL-C < 70 mg/dL, non-HDL-C < 85 mg/dL) were established on the basis of the guidelines in place in 2015 [14,15,16]. The analyses were extended to include the therapeutic goals proposed in relevant guidelines that appeared in subsequent years, i.e., the guidelines on hypertension (2018) [20] and dyslipidaemia (2019) [21].

### 2.2. Statistical Analyses

Mean values, standard deviations, proportions of risk factors, and treatment used are presented and compared between the patients from urban and rural areas using Pearson’s chi-square or Mann–Whitney U or Student’s *t*-tests, as appropriate. The relationship between the patients’ characteristics and their achievement of lipid-lowering goals, hypertension, and diabetes therapy was examined separately for urban and rural areas, and each variable was analysed individually. Logistic regression was used to investigate the relationship between all independent variables and achievement of: (1) lipid-fraction target levels, (2) blood-pressure target levels, (3) glucose and HbA1c target levels.

Odds ratios (OR) and 95% confidence intervals (CI) are presented. *P*-values below 0.05 were considered significant. All statistical analyses were conducted using the STATISTICA programme, version 13.3.

## 3. Results

### 3.1. Patient Characteristics

The LIPIDOGRAM2015 study included 13,724 patients (7028 residents of urban and 6696 rural areas, respectively). The average age was 56 years (57 in urban and 56 in rural populations; *p* < 0.01). The average age of women living in cities was 2 years older than those living in rural settings (*p* < 0.01). The level of education was higher in urban areas in both genders (*p* < 0.01). Demographic characteristics of enrolled patients (categorised by gender and place of residence) are presented in Table 1.

### 3.2. Risk factors and Cardiovascular Diseases

There were no statistical differences between the mean levels of total cholesterol, triglycerides, glucose, and HbA1c in patients living in urban and rural areas. Mean systolic and diastolic BP was 1 mmHg higher in the rural residents (a statistically significant difference). HDL-C concentration was statistically higher in the urban female population and non-HDL-C level in the urban male population. The prevalence of hypertension, DM, dyslipidaemia, and CVD was similar in urban and rural areas. The prevalence of obesity and excessive waist circumference (77.49% vs. 80.66%, *p* < 0.01), as well as abdominal obesity (43.18 vs. 46.40%, *p* < 0.01), were higher in rural areas in both genders. However, there was no such trend in the overweight category—in men, the prevalence was higher in the urban population, while in women, the differences were not statistically significant. In addition, women living in rural areas smoked less frequently than those in urban areas (a statistically significant difference), while the differences in smoking status in men were not statistically significant. Detailed data are presented in Table 1.

There were also no statistically significant differences in the use of lipid-lowering therapies, antihypertensives, or most diabetic medications between patients living in urban and rural areas. The most frequently used lipid-lowering medicine were: atorvastatin, simvastatin, and rosuvastatin. Diabetes was treated mostly with metformin. The groups of antihypertensive drugs most commonly used in PHC practices were ACEI, diuretics, and B-blockers. Glipizide was significantly more frequently used in the female rural population. Detailed values are presented in Appendix A.

### 3.3. Achievement of Therapeutic Goals

A proportion of 40% of patients (both in urban and rural areas) without diagnosed diabetes had elevated fasting glucose levels. More than 55% of diabetic patients achieved HbA1c < 7% (Figure 1).

Roughly three quarters of patients without diagnosed hypertension had BP below 140/90 mmHg measured in the office. Forty-eight per cent of hypertensive patients in the urban population compared to 43% in the rural population achieved recommended BP targets (according to guidelines in force at the time the study was conducted) (*p* < 0.01). The differences were statistically significant in both genders (Figure 2). After analysing data in light of BP target values introduced by the 2018 ESH/ESC guidelines [20] (published after the study period), we discovered that new treatment targets were achieved in approximately 5% of patients. Detailed data are presented in Appendix A.

Over 36% of all patients without CVD had LDL-C < 115 mg/dL. In this group, around 22% had LDL-C < 100 mg/dL. Almost half of the patients reached non-HDL-C < 145 mg/dL, and around 2/3 of patients had TG < 150 mg/dL. The differences between urban and rural populations were not statistically significant (Figure 3). Fourteen per cent of patients with CVD from urban areas and 11% from rural areas reached LDL < 70 mg/dL (*p* = 0.04). The differences between urban and rural populations were statistically significant only in males (18% in urban areas vs. 12% in rural areas, *p* < 0.01). There were also statistically significant differences between the percentage of patients with CVD reaching therapeutic goals of non-HDL-C in the urban and rural male population (11.21 vs. 7.32%, *p* = 0.04) (Figure 3).

Applying the lower LDL-C target values introduced by the 2019 ESC/EAS guidelines [21] (published after the study period), treatment targets were achieved in less than 5% of patients. Detailed data are presented in Appendix A.

Patients with higher education were more likely to achieve therapeutic levels of LDL-C, BP, and glucose both in urban and rural populations (OR = 1.24, OR = 2.68, OR = 2.04, respectively). Diabetic patients living in urban areas had a slightly higher probability of achieving therapeutic LDL-C levels (OR = 1.15 in urban and OR = 1.06 in rural population). Age, male gender, co-existence of CVD risk factors, and other diseases were associated with a lower likelihood of reaching therapeutic levels of LDL-C, BP, and glucose. For both urban and rural populations, the strongest negative association with reaching therapeutic LDL-C goals was the co-existence of CVD (OR = 0.17 and OR = 0.13, respectively), while the presence of obesity was the strongest factor associated with failing to reach goals for BP and glucose levels. Detailed data are presented in Table 2.

## 4. Discussion

### 4.1. Main Findings

We found no important differences in the prevalence of hypertension, DM, dyslipidaemia, or CVD in urban and rural adult populations attending primary care in Poland. Nor did we find differences in mean levels of glucose or HbA1c. However, obesity was more common among rural patients. There were no significant differences in the use of lipid-lowering, hypertensive, and diabetic medications between patients living in urban and rural areas.

Both urban and rural women achieved recommended blood-pressure targets more often than men. Women without established CVD were also more likely to achieve their target LDL-C, non-HDL-C, and TG values than men.

In urban areas, more patients with hypertension achieved their therapeutic goals. The same tendency was observed in patients with CVD attaining recommended LDL-C and non-HDL-C targets in the male population.

Older age, male gender, co-existence of CV risk factors (e.g., excessive weight or waist circumference, smoking, dyslipidaemia, hypertension, diabetes), and CVD, were associated with a lower likelihood of reaching therapeutic levels of LDL-C, BP, and glucose both in urban and rural populations. Of these factors, diagnosed CVD was most strongly associated with failure to reach therapeutic LDL-C goals. Obesity was strongly negatively associated with the achievement of therapeutic goals for BP and glucose.

### 4.2. Comparison with Other Studies

Poland is classified as a country with high CVD mortality [22]. Furthermore, several studies have observed high CVD morbidity and a high prevalence of CV risk factors in Poland.

The NATPOL2011 study, which was conducted in Poland in 2011, showed that the prevalence of hypercholesterolaemia was 63.6% in villages, 55% in towns, with 50,000–200,000 citizens, and 59.5% in towns with >200,000 citizens [23]. The WOBASZ study measured the prevalence, in big Polish cities, of CVD risk factors such as overweight (47.9% in men, 39.0% in women), obesity (23.3% in men, 26.8% in women), HTN (68.8% in men, 58.3% in women), hypercholesterolemia (82.6% in men, 87.4% in women), and DM (10.4% in men, 8.6% in women) [24]. In our study, the prevalence of obesity, dyslipidaemia, and diabetes was higher since the population of LIPIDOGRAM2015 does not represent the general population of Poland, like the studies mentioned above, but those attending PHC practices.

In 2018, Nowicki et al. published the results of a study conducted on 4040 inhabitants of Eastern Poland, aged 35–64, without a prior history of CV events [25]. Residents of urban areas more frequently achieved recommended levels of BMI (27 vs. 22%), BP (12 vs. 9%), fasting cholesterol (38 vs. 35%), and fasting glucose (66 vs. 59%) than people living in rural areas. The prevalence of obesity was higher in rural areas (31 vs. 38%). Compared with our study, greater differences between the city and the countryside can be explained by the method of recruiting patients (population selected from the electoral record) and the fact that the study was conducted in a less-developed region than the Polish average.

There are also some differences in the prevalence of CVD and CV risk factors between urban and rural populations in European countries and the United States.

In 2011, Markowić et al. published a study of 2467 participants ≥40 years old, who visited their general practitioner for any reason. The study included 59 general practices, covering all of Croatia [6]. Participants from rural areas had significantly higher systolic and diastolic BP, while no clinically relevant differences in cholesterol fraction levels (TC, LDL-C, HDL-C, and TG) were observed. Obesity was more common in rural patient populations, both in the continental (35 vs. 45%) and coastal (32 vs. 36%) areas. 

Lindroth et al. showed that the overweight and obesity rates were higher in rural than urban areas (43 vs. 37% and 26 vs. 19%, respectively). The differences in SBP, DBP, and fasting glucose levels were not statistically significant. Cholesterol levels were lower in cities than in the countryside, although the difference was minor (5.4 vs. 5.8 mmol/L in males and 5.4 vs. 5.8 mmol/L in females) [7].

The U.S. Atherosclerosis Risk in Communities Study (ARIC) included white persons aged 45–64 (*n* = 7497) selected in urban and semi-rural areas surveyed in 1987–1989. The TC and BP levels in both populations were similar [26,27].

Overall, the most recent studies from various developed countries have reported similar results to ours and have found minor differences in CVD risk factors between rural and urban areas.

### 4.3. Interpretation of Study Results

The results of our study confirm previous observations in relation to the high prevalence of CVD and CV risk factors in the Polish population. This is true in both urban and rural areas. Although there are differences in the prevalence of CV risk factors between urban and rural populations in Poland, these differences in BP, cholesterol fractions, and glucose levels are small. Nevertheless, these differences should be taken into account from the public health point of view. They may be explained by differences in patients’ living conditions and lifestyles. Differences may also reflect disparities in access to healthcare, as described in other European studies [6,7]. Although we did not analyse these factors, they should be taken into account when interpreting our findings.

Our study results showed that less than half of hypertensive patients and less than 10% of patients with CVD are reaching therapeutic goals for LDL-C. Those values are far from satisfactory and indicate a clear need for improvements in care. Pharmacotherapy used in the treatment of CVD, hypertension, and diabetes is similar in Polish urban and rural areas.

### 4.4. Implications of Study Results

The results of our study may be important for family doctors and their patients. Only around 50% of primary care patients in Poland achieve recommended targets for BP, glucose, and HbA1c. Of particular concern is the very low percentage of patients with CVD reaching the therapeutic level of LDL-C. Thus, it is necessary to put more emphasis on (1) understanding disease states and their implications for long-term health, (2) educating patients about the efficacy and importance of any drugs they are prescribed, and (3) emphasizing the importance of long-term adherence to drugs, lifestyle modification, and clinical follow-up in order to optimize health care and management of CV risk factors. Concerning lipids, it is essential to encourage physicians to implement more effective management of patients, focusing on lifestyle changes and, when required, the use of combination therapy and share care arrangements with other specialists. Since the use of ezetimibe is suboptimal, PHC physicians should put more emphasis on combined statin–ezetimibe therapy. Particular attention should be paid to older men with a low level of education and obese patients, especially in rural areas. We are currently seeing a trend towards lowering the target values of cholesterol and blood pressure (introduced in 2018 and 2019 by ESC guidelines) [20,21]. After analysing our data in light of those new target values, we see the further need to implement more effective management of patients.

The results of our study are of interest to public health specialists. The high prevalence of CV risk factors and problems with achieving therapeutic goals indicate that the existing care and implemented national prevention programs are insufficient. The results of our study indicate that patients from rural areas likely have unrecognized obstacles to achieving target levels of BP, LDL-C, and non-HDL-C. It seems advisable to develop a patient information program and an educational agenda aimed at medical staff to improve the situation.

Researchers should plan and perform additional systematic studies on CVD and CV risk factor prevalence in the Polish PHC patient population. Such follow-up studies will help to assess whether the situation is changing in the desired direction.

## 5. Conclusions

The prevalence of CVD and CV risk factors among urban and rural adult populations attending primary care in Poland is high. There were no important differences in the prevalence of hypertension, diabetes, dyslipidaemia, and CVD between those groups, while obesity was found to be slightly more common in rural areas. Patients with CVD, in particular, have difficulties in achieving LDL-C target goals, while the rural male population with hypertension has less well-controlled BP. Therefore, there is a need to implement CVD prevention programs in Polish primary care aimed both at urban and rural areas residents, with greater attention paid to the rural population.

## Figures and Tables

**Figure 1 jcm-10-05656-f001:**
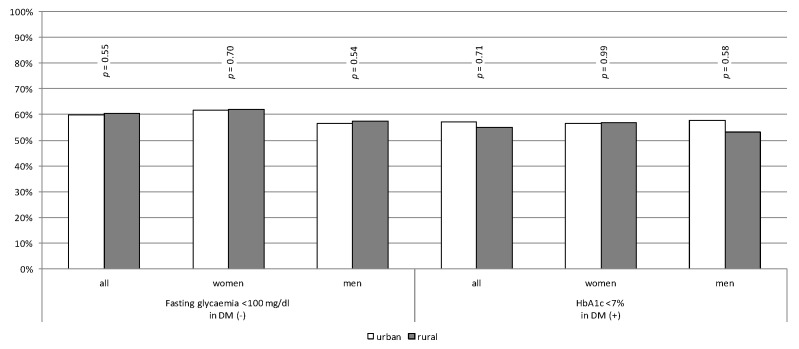
Percentage of patients without diabetes with normal fasting glucose level and diabetic patients reaching treatment goals.

**Figure 2 jcm-10-05656-f002:**
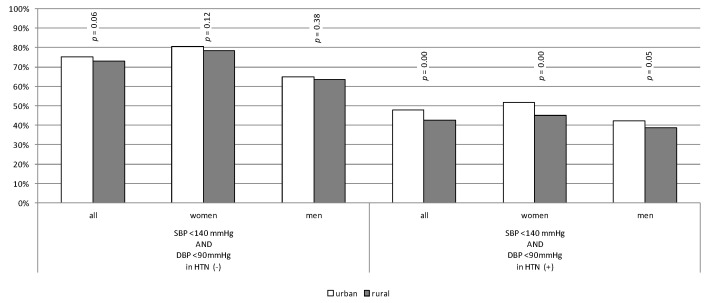
Percentage of patients without hypertension with normal blood-pressure level and hypertensive patients reaching treatment goals.

**Figure 3 jcm-10-05656-f003:**
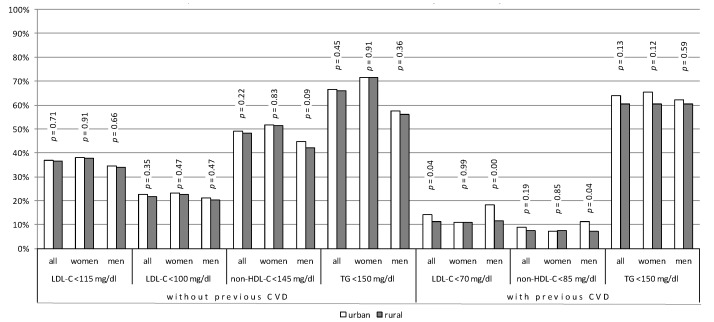
Percentage of patients without cardiovascular diseases with normal cholesterol-fractions level and patients with cardiovascular diseases reaching treatment goals.

**Table 1 jcm-10-05656-t001:** Participant clinical characteristics.

	All			Women			Men		
	Urban	Rural	*p*	Urban	Rural	*p*	Urban	Rural	*p*
Total *n* (%)	7028 (51.21%)	6696 (48.79%)	0.11	4495 (51.7%)	4195 (48.3%)	0.11	2533 (50.3%)	2501 (49.7%)	0.11
$\mathrm{Age}\;\bar{X}\pm SD$ (years)	57 ± 14	56 ± 14	0.00	57 ± 14	55 ± 14	0.00	56 ± 14	56 ± 14	0.17
Levels of education *n* (%)									
Primary	546 (7.77%)	1181 (17.64%)		397 (8.83%)	796 (18.97%)		149 (5.88)	385 (15.39)	
Vocational	1754 (24.96%)	2298 (34.32%)		918 (20.42%)	1166 (27.79%)		836 (33.0%)	1132 (45.26%)	
Secondary	2811 (40.00%)	3112 (31.54%)		1931 (42.96%)	1448 (34.52%)		880 (34.74%)	664 (26.55%)	
High	1917 (27.28%)	1105 (16.50%)	0.00	1249 (27.79%)	785 (18.71%)	0.00	668 (26.37%)	320 (12.79%)	0.00
SBP $\bar{X}\pm SD$ (mmHg)	132 ± 18	133 ± 19	0.01	130 ± 18	131 ± 19	0.05	136 ± 19	135 ± 17	0.14
DBP $\bar{X}\pm SD$ (mmHg)	80 ± 10	81 ± 11	0.00	79 ± 10	80 ± 11	0.00	83 ± 11	82 ± 10	0.03
$\mathrm{TC}\;\bar{X}\pm SD$ (mg/dL)	202 ± 44	203 ± 45	0.21	205 ± 44	205 ± 43	0.92	199 ± 47	196 ± 44	0.02
$\mathrm{HDL}-C\;\bar{X}\pm SD$ (mg/dL)	55 ± 15	55 ± 15	0.02	59 ± 15	58 ± 15	0.00	48 ± 13	48 ± 13	0.14
$\mathrm{LDL}-C\;\bar{X}\pm SD$ (mg/dL)	128 ± 40	130 ± 41	0.04	129 ± 41	130 ± 41	0.23	129 ± 40	126 ± 39	0.05
$\mathrm{non}-\mathrm{HDL}-C\;\bar{X}\pm SD$ (mg/dL)	147 ± 42	148 ± 43	0.03	146 ± 42	147 ± 41	0.24	151 ± 45	148 ± 42	0.04
$\mathrm{TG}\;\bar{X}\pm SD$ (mg/dL)	147 ± 113	149 ± 124	0.79	135 ± 97	135 ± 83	0.83	174 ± 170	169 ± 133	0.78
$\mathrm{Glycaemia}\;\bar{X}\pm SD$ (mg/dL)	104 ± 25	103 ± 25	0.048	102 ± 24	102 ± 23	0.07	106 ± 27	106 ± 27	0.27
$\mathrm{HbA}1c\;\bar{X}\pm SD$ (%)	5.7 ± 1	5.7 ± 1	0.61	5.7 ± 0.9	5.7 ± 0.9	0.83	5.8 ± 1	5.8 ± 1	0.39
Hypertension *n* (%)	3482 (49.54%)	3306 (49.37%)	0.84	2152 (47.88%)	1990 (47.44%)	0.68	1330 (52.51%)	1316 (52.62%)	0.94
Diabetes *n* (%)	964 (13.72%)	879 (13.13%)	0.31	534 (11.88%)	487 (11.61%)	0.70	430 (16.98%)	392 (15.67%)	0.21
Dyslipidaemia *n* (%)	5917 (84.22%)	5706 (85.22%)	0.11	3698 (82.27%)	461 (82.50%)	0.77	2221 (87.68%)	2245 (89.76%)	0.02
CVD *n* (%)	1014 (14.43%)	951 (14.20%)	0.70	550 (12.24%)	459 (10.94%)	0.06	464 (18.32%)	492 (19.67%)	0.22
CHD	921 (13.10%)	866 (12.93%)	0.77	492 (10.95%)	415 (9.89%)	0.11	429 (16.94%)	451 (18.03%)	0.31
MI	335 (4.77%)	290 (4.33%)	0.22	113 (2.51%)	81 (1.93%)	0.07	222 (8.76%)	209 (8.36%)	0.61
Ischaemic stroke	145 (2.06%)	123 (1.84%)	0.34	80 (1.78%)	61 (1.45%)	0.23	65 (2.57%)	62 (2.48%)	0.84
Haemorrhagic stroke	17 (0.24%)	14 (0.21%)	0.69	8 (0.18%)	8 (0.19%)	0.89	9 (0.36%)	6 (0.24%)	0.45
AF	343 (4.88%)	365 (5.45%)	0.13	208 (4.63%)	202 (4.82%)	0.68	135 (5.33%)	163 (6.52%)	0.07
FH	32 (0.46%)	41 (0.61%)	0.21	25 (0.56%)	27 (0.64%)	0.60	7 (0.28%)	14 (0.56%)	0.12
Overweight *n* (%)	2822 (40.15%)	2595 (38.75%)	0.09	1623 (36.11%)	1528 (36.42%)	0.76	1199 (47.34%)	1067 (42.66%)	0.00
Obesity *n* (%)	2273 (32.34%)	2513 (37.53%)	0.00	1365 (30.37%)	1497 (35.69%)	0.00	908 (35.85%)	1016 (40.62%)	0.00
Exceeded waist circumference (W ≥ 80 cm; M ≥ 94 cm) *n* (%)	5446 (77.49%)	5401 (80.66%)	0.00	3563 (79.27%)	3478 (82.91%)	0.00	650 (74.34%)	578 (76.89%)	0.04
Abdominal obesity *n* (%)	3035 (43.18%)	3107 (46.40%)	0.00	2292 (50.99%)	2356 (56.16%)	0.00	743 (29.33%)	751 (30.03%)	0.59
Smoking status *n* (%)									
Never smokers	3648 (51.91%)	3742 (55.88%)		2620 (58.29%)	2781 (66.29%)		1028 (40.58%)	961 (38.42%)	
Current smokers	1247 (17.74%)	1056 (15.77%)		748 (16.64%)	551 (13.13%)		499 (19.70%)	505 (20.19%)	
Ex-smokers	2133 (30.35%)	1898 (28.35%)	0.00	1127 (25.07%)	863 (20.57%)	0.00	1006 (39.72%)	1035 (41.38%)	0.29

*p* < 0.05 was considered statistically significant. SD: standard deviation, SBP: systolic blood pressure, DBP: diastolic blood pressure, TC: total cholesterol, HDL-C: high-density lipoprotein cholesterol, LDL-C: low-density lipoprotein cholesterol, TG: triglycerides, HbA1c: glycated haemoglobin, CVD: cardiovascular disease, CHD: coronary heart disease, MI: myocardial infarct, AF: atrial fibrillation. To receive values in mmol/L: (1) for TC, HDL-C, and LDL-C divide the value in mg/dL by 38.67; (2) for TG divide the value in mg/dL by 88.57.

**Table 2 jcm-10-05656-t002:** Logistic regression analysis of factors associated with the achievement of goals in the lipid-lowering, hypertension, and diabetes therapy, among study patients in urban and rural area.

	Lipids				Blood Pressure				Glucose			
	Urban		Rural		Urban		Rural		Urban		Rural	
Analysed Variables	OR (95% CI)	*p*	OR (95% CI)	*p*	OR (95% CI)	*p*	OR (95% CI)	*p*	OR (95% CI)	*p*	OR (95% CI)	*p*
Age	0.98 (0.98–0.98)	0.00	0.98 (0.98–0.98)	0.00	0.97 (0.97–0.97)	0.00	0.96 (0.96–0.96)	0.00	0.97 (0.97–0.98)	0.00	0.97 (0.97–0.98)	0.00
Male	0.83 (0.75–0.91)	0.00	0.71 (0.65–0.79)	0.00	0.57 (0.51–0.63)	0.00	0.60 (0.54–0.67)	0.00	0.80 (0.73–0.88)	0.00	0.80 (0.72–0.88)	0.00
Levels of education												
Vocational	0.89 (0.74–1.09)	0.26	0.90 (0.78–1.03)	0.13	1.16 (0.96–1.41)	0.13	1.37 (1.19–1.57)	0.00	1.16 (0.96–1.41)	0.12	1.23 (1.07–1.41)	0.00
Secondary	0.99 (0.83–1.19)	0.94	1.06 (0.92–1.22)	0.42	1.59 (1.32–1.92)	0.00	2.21 (1.91–2.55)	0.00	1.51 (1.25–1.81)	0.00	1.67 (1.44–1.93)	0.00
High	1.24 (1.02–1.50)	0.03	1.39 (1.18–1.64)	0.00	2.68 (2.20–3.26)	0.00	3.56 (2.98–4.24)	0.00	2.04 (1.68–2.47)	0.00	2.19 (1.85–2.60)	0.00
Hypertension	0.77 (0.70–0.84)	0.00	0.84 (0.76–0.92)	0.00	0.31 (0.28–0.34)	0.00	0.27 (0.24–0.30)	0.00	0.50 (0.46–0.55)	0.00	0.53 (0.48–0.58)	0.00
Diabetes	1.15 (1.00–1.32)	0.04	1.06 (0.92–1.23)	0.39	0.47 (0.41–0.54)	0.00	0.46 (0.40–0.53)	0.00	0.69 (0.60–0.79)	0.00	0.61 (0.53–0.71)	0.00
Dyslipidaemia	0.73 (0.66–0.80)	0.00	0.76 (0.69–0.84)	0.00	0.70 (0.64–0.77)	0.00	0.64 (0.58–0.71)	0.00	0.64 (0.58–0.70)	0.00	0.69 (0.63–0.76)	0.00
CVD	0.17 (0.14–0.20)	0.00	0.13 (0.11–0.16)	0.00	0.58 (0.51–0.66)	0.00	0.59 (0.51–0.67)	0.00	0.65 (0.57–0.75)	0.00	0.60 (0.52–0.69)	0.00
Overweight	0.62 (0.55–0.70)	0.00	0.63 (0.56–0.72)	0.00	0.48 (0.42–0.55)	0.00	0.51 (0.45–0.59)	0.00	0.59 (0.52–0.67)	0.00	0.56 (0.48–0.64)	0.00
Obesity	0.59 (0.52–0.67)	0.00	0.60 (0.53–0.69)	0.00	0.28 (0.25–0.32)	0.00	0.28 (0.25–0.32)	0.00	0.38 (0.34–0.44)	0.00	0.33 (0.29–0.38)	0.00
Exceeded waist circumference (W ≥ 80 cm; M ≥ 94 cm)	0.63 (0.56–0.71)	0.00	0.64 (0.57–0.72)	0.00	0.43 (0.38–0.49)	0.00	0.39 (0.34–0.44)	0.00	0.45 (0.40–0.51)	0.00	0.45 (0.39–0.51)	0.00
Abdominal obesity	0.81 (0.74–0.90)	0.00	0.86 (0.78–0.95)	0.00	0.67 (0.60–0.73)	0.00	0.66 (0.60–0.73)	0.00	0.64 (0.58–0.70)	0.00	0.65 (0.59–0.72)	0.00
Smoking status												
Ex-smokers	0.85 (0.76–0.95)	0.00	0.82 (0.74–0.92)	0.00	0.78 (0.70–0.87)	0.00	0.90 (0.80–1.00)	0.06	0.78 (0.70–0.86)	0.00	0.86 (0.77–0.96)	0.01
Current smokers	0.80 (0.70–0.91)	0.00	0.79 (0.68–0.90)	0.00	0.99 (0.87–1.13)	0.90	1.02 (0.89–1.18)	0.73	0.85 (0.74–0.97)	0.01	0.95 (0.83–1.10)	0.50

*p* < 0.05 was considered statistically significant. OR: odds ratio; CI: confidence interval; CVD: cardiovascular disease.

## Data Availability

The data underlying this article will be shared on reasonable request to the corresponding author.

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
