# Peer review of "The Differences in the Prevalence of Cardiovascular Disease, Its Risk Factors, and Achievement of Therapeutic Goals among Urban and Rural Primary Care Patients in Poland: Results from the LIPIDOGRAM 2015 Study"

_jcm, 2021, doi:10.3390/jcm10235656_

Round 1

Reviewer 1 Report

I've read with attention the paper entitled  "The differences in the prevalence of cardiovascular disease, its risk factors and achievement of therapeutic goals among urban and rural primary care patients in Poland: results from the LIPIDOGRAM 2015 study"  that is potentially of interest.

The methodology applied is overall correct, and the results are reliable.

There are some additional questions that I whould like to ask:

Why is dyslipidaemia defined as a level greater or equal than 115mg/dl?

Should be better explained with refferences, excessive waist circumference in abstract and also in main body text. Should be defined more detailed what formula was used for visceral body fat calculations.

Reviewer 2 Report

The authors present a secondary analysis of data from the nation-wide cross-sectional study in primary care patients in Poland (2015). The study concerns the prevalence of cardiovascular disease and its risk factors in urban and rural settings, which is an important public health issue.

Major comments.

The manuscript can be improved by more explicit presentation of the cardiovascualr risk factors that can be modified by clinical or behavioural interventions based on the current guidelines.  Thus, the authors use different definitions of obesity that are based on BMI, waist circumference and WHR (abdominal obesity). Do these parameters have the same clinical significance according to the current European guidelines? Please re-check the recommendations of the European guidelines on prevention of cardiovascular disease in clinical practice.   

 Discussion section would benefit by comparing your study results with the studies in similar populations (PHC settings). Gender-related differences in achievement of therapeutic targets in rural/urban areas could be paid more attention. 

Other comments. Please find in the pdf.file with comments.

Ethical considerations. If this study is an exempt from the ethical committee review, please state that in the methods or in a separate subsection.

Round 2

Reviewer 2 Report

The previous comments were mostly addressed. 

The definition of ex-smokers is still inappropriate. What question was used in the study questionnaire to differentiate/define the smoking status?  Please re-visit the questionnaire and adjust the categories. If you had more than one question, still you have to make more clear who were considered to be ex-smokers. For example, those who smoked for certain period of time but not on the study inclusion day.

"Patients who smoked at least one cigarette per day on the day of inclusion in the study were classed as current smokers. Patients who smoked cigarettes solely before that day were classed as ex-smokers. Patients who never smoked cigarettes were classed as non-smokers."